# Targeted Treatment of Soft-Tissue Sarcoma

**DOI:** 10.3390/jpm13050730

**Published:** 2023-04-26

**Authors:** Anne Iren Riskjell, Vivi-Nelli Mäkinen, Birgitte Sandfeld-Paulsen, Ninna Aggerholm-Pedersen

**Affiliations:** 1Faculty of Health, Aarhus University, 8000 Aarhus C, Denmark; annrij@rm.dk (A.I.R.); vivmaa@rm.dk (V.-N.M.); 2Department of Clinical Biochemistry, Viborg Regional Hospital, 8800 Viborg, Denmark; 3Department of Oncology, Aarhus University Hospital, Palle Juul-Jensens Boulevard 99, 8200 Aarhus N, Denmark

**Keywords:** targeted treatment, gene mutations, sarcoma

## Abstract

**Background**: Soft-tissue sarcoma (STS) is a heterogeneous group of sarcomas with a low incidence. The treatment of advanced disease is poor, and mortality is high. We aimed to generate an overview of the clinical experiences with targeted treatments based on a pre-specified target in patients with STS. **Methods**: A systematic literature search was conducted in PubMed and Embase databases. The programs ENDNOTE and COVIDENCE were used for data management. The literature was screened to assess the article’s eligibility for inclusion. **Results**: Twenty-eight targeted agents were used to treat 80 patients with advanced STS and a known pre-specified genetic alteration. MDM2 inhibitors were the most-studied drug (*n* = 19), followed by crizotinib (*n* = 9), ceritinib (*n* = 8), and ^90^Y-OTSA (*n* = 8). All patients treated with the MDM2 inhibitor achieved a treatment response of stable disease (SD) or better with a treatment duration of 4 to 83 months. For the remaining drugs, a more mixed response was observed. The evidence is low because most studies were case reports or cohort studies, where only a few STS patients were included. **Conclusions**: Many targeted agents can precisely target specific genetic alterations in advanced STS. The MDM2 inhibitor has shown promising results.

## 1. Introduction

Sarcomas are neoplasms originating from the connective tissue. With more than eighty histological subtypes, sarcomas are highly heterogenous and rare, representing only about 1–2% of all adult cancers [1,2]. Sarcomas are divided into soft-tissue sarcoma (STS), representing approximately 84% of sarcoma patients, and sarcomas from bone and cartilage, representing approximately 16% of the patients [3]. The standard treatment of localised disease is surgery, with or without radiotherapy [4]. Despite a curative-intended therapy, the prognosis is grave as 25% of the patients will develop metastatic disease. These patients are primarily treated with chemotherapy [5,6]. However, the response to chemotherapy is low, and the mortality is high [2]. Therefore, the need for improved treatment for patients with advanced STS is needed.

Over the past decades, multiple genetic alterations in cancer have been identified. This has revolutionised the treatment by enabling targeted treatment to precisely inhibit the growth and progression of cancer cells [7].

STS is driven primarily by a fusion of genes rather than mutations. However, several studies have shown some genetic alterations in STS. The Cancer Genome Atlas (TCGA) described the molecular landscape of 206 adult STS, representing six subtypes of STS, including synovial sarcoma (SS), liposarcoma (LPS), leiomyosarcoma (LMS), malignant peripheral nerve sheath sarcoma (MPNST), myxofibrosarcoma, and undifferentiated sarcoma. They found that most sarcomas are characterised by copy number alterations (CNAs) and a low tumour mutation burden (TMB) [8]. TMB is usually categorised into three categories: low (1–5 mutations/Mb), intermediate (6–19 mutations/Mb), and high (≥20 mutations/Mb) [9]. The average TMB in the TCGA was low for sarcoma, with an average of 1.06 mutations/Mb. Other mutations represent a few highly recurrent genes. The most frequently mutated genes in the database were TP53 (*n* = 69), ATRX (*n* = 31), RB1 (*n* = 18), PCLO (*n* = 9), FAT1 (*n* = 6), NF1 (*n* = 6), PRKDC (*n* = 6), and LRP1B (*n* = 6). The most frequent amplifications were seen in MDM2 (*n* = 46), FRS2 (*n* = 45), CDK4 (*n* = 44), HMG2A (*n* = 36), and PTPRB (*n* = 33). The most frequent deletions were observed in CDKN2A (*n* = 22), CDKN2B (*n* = 22), RB1 (*n* = 22), CYSLTR2 (*n* = 18), and TP53 (*n* = 16) [8]. In a study investigating the mismatch repair (MMR) status of 304 sarcomas, seven were found to be MMR-deficient. MMR-deficient sarcomas had a significantly higher TMB than MMR-proficient ones, with an average of 16.95 mutations/Mb and 4.56 mutations/Mb, respectively [10].

Arnaud-Coffin et al. investigated 158 patients with advanced STS by genetic profiling. They found 289 relevant genetic alterations in 149 genes. The most frequent alterations were TP53 (*n* = 36), RB1 (*n* = 22), CDKN2A (*n* = 17), CDK4 (*n* = 9), MDM2 (*n* = 8), and PTEN (*n* = 7). The alterations in CDK4 and MDM2 were amplifications only [2]. In a study including 102 sarcoma patients, the most frequent alterations were also TP53 (*n* = 32), CDK4 (*n* = 24), MDM2 (*n* = 22), RB1 (*n* = 19), and CDKN2A/B (*n* = 14) [11]. In a study investigating the molecular characterisation of fourteen adult STS, the most frequently altered genes were FRGB1 (*n* = 8) and CDC27 (*n* = 6). TP53, ARTX, and PTEN were mutated in three cases. The median TMB was low (2.38 mutations/Mb) [12]. Dembla et al. found the prevalence of MDM2 amplification in 13/33 sarcoma patients [13]. Lin et al. included 301 with uterine sarcomas representing many subtypes of STS. Here, they found that nineteen were SMARC4-deficient. In the SMARC4-deficient cohort, they performed next-generation sequencing (NGS) on sixteen patients. The average TMB was low (1.7 mutations/Mb), and they found mutations in TP53 (*n* = 2), RB1 (*n* = 1), CTNNB1 (*n* = 1), and ZNF703 (*n* = 1) [14]. Seol et al. also analysed five patients with uterine sarcoma with NGS. In one patient, they found amplifications of AKT3, BRAF, and EGFR. Another patient had an amplification of PDGFRB [15]. Li et al. investigated forty BCOR-rearranged uterine sarcomas. Among these patients, thirty-eight had ZC3H7B–BCOR fusion. They also found amplifications of MDM2 (*n* = 18), FRS2 (*n* = 16), CKD4 (*n* = 15), PDGFRA (*n* = 3), KDR (*n* = 2), ERBB3 (*n* = 2), and KIT (*n* = 1). Loss of CDKN2A and CDKN2B were observed in eleven and seven cases, respectively. They also found inactivating mutations in TP53 (*n* = 4), PTCH1 (*n* = 2), NF1 (*n* = 2), and NF2 (*n* = 1). In addition to BCOR, other rearrangements were in the HMGA2 and NCOR2 genes, with six and two cases, respectively. Thirty-nine patients had a low TMB, and one had an intermediate. In addition to the BCOR cohort, they also investigated a cohort consisting of fifteen patients with BCOR internal tandem duplication. Here, none of the cases had MDM2 and CDK4 amplifications. Three of the cases had a CDKN2A/B loss and mutations in STAG2 (*n* = 2), PASK (*n* = 2), and ARID1A (*n* = 2) [16]. Thirteen patients with renal sarcoma had a low TMB with a mean of 3.5 mutations/Mb. Amplifications of KIT and PDGFRA were observed in four patients. Three cases experienced a loss of CDKN2A/B. Genetic alterations were also found in TP53 (*n* = 4), NF1 (*n* = 3), and MLL2 (*n* = 2). One case had a fusion of STAT6–NAB2 [17].

In STS, the frequency of genetic alterations is between 84 and 91%, with the most frequently altered genes being TP53, ATRX, RB1, PTEN, MDM2, CDK4, and CDKN2A/B [2,8,11,13,16,17,18,19]. Different genetic alterations can enable targeted therapy in patients suffering from advanced STS. However, while much is currently known about the genetic landscape of sarcoma, no overview covering treatment options against specific genetic alterations in STS exists. This systematic review aims to generate an overview of the clinical experiences with targeted treatments based on a pre-specified target in patients with STS. This study focuses on the outcome after targeted treatment of rare genetic alterations in sarcoma patients, not the genetic alteration by themselves or already effective treatments. Genetic testing is primarily performed in sarcoma patients with locally advanced or metastatic diseases treated with known standard palliative treatment. The studies included in this systematic review were selected for treatment given based on the results from comprehensive genetic testing.

## 2. Materials and Methods

### 2.1. Data Sources and Search Strings

A systematic review of the existing literature investigating the targeted treatment of adult STS was performed according to the Preferred Reporting Items for Systematic Reviews and Meta-analysis (PRISMA) guidelines [20]. A comprehensive literature search in the medical databases PubMed and Embase was conducted on 1 April 2022. The following filters were used in both databases: “not animals” and “English”. There was no time restriction. The search strings used are presented in Table 1.

### 2.2. Study Selection

The inclusion criteria were as follows: (I) original data, (II) the patients had to have a proven pathological STS with a specific genetic alteration prior to therapy with a targeted drug, (III) treatment outcome of one or several targeted agents had to be presented, (IV) if a study also included other types of cancer, the treatment outcome on STS had to be presented separately, (V) the patients had to be older than 15 years old, (VI) if a study included both paediatric and adult STS, the treatment outcome regarding adult STS had to be presented separately. Exclusion criteria were as follows: (I) non-English articles, (II) conference abstracts, (III) animal or in vitro studies, (IV) bone or cartilage sarcomas, hemopoietic sarcomas, and sarcomatoid tumours, (V) results only presented in figures, (VI) studies regarding immunotherapy. Patients with gastrointestinal stomal tumours (GISTs) treated with tyrosine kinase inhibitors (TKIs) were not included in the systemic review, even though they are treated with TKIs targeting different tyrosine kinases. The evidence for this treatment is well known, and for a systematic review of treating GIST, we referred to a systematic review published by Brinch et al. [21]. The multi-targeted drug pazopanib, a tyrosine kinase inhibitor targeting VEGFR, a platelet-derived growth factor receptor, and a c-kit were not included, as the treatment is given to many sarcoma patients with genetic testing with good clinical responses [22,23,24,25,26,27,28]. The same is true for regorafenib [29,30].

### 2.3. Data Extraction and Quality Assessment

All titles and abstracts were screened to identify eligible articles. One hundred studies were randomly selected and screened independently by all four authors to validate the abovementioned in- and exclusion criteria. Any disagreement was resolved by consensus. Two of the authors (AIR and VNM) screened the rest of the titles and abstracts. In the case of disagreement, conflicts were resolved by all four authors.

After the identification of eligible full texts, ten randomly selected articles were read by three authors (AIR, BSP, and NAP) and subsequently included or excluded. The rest were included or excluded through full-text reading by one author (AIR) and, in the case of doubt, discussed by the other authors until a consensus was reached. Covidence (covidence.org) and Endnote (Clarivate Analytics) were used for duplicate and reference management during the inclusion and exclusion processes. The protocol was submitted to the PROSPERO database (CRD42021252341).

One author (AIR) performed data extraction, which subsequently was checked by the other authors. Data extraction included the first author’s name, year of publishing, study design, genetic alteration that served as a target, type and/or name of the targeted drug, population, and treatment outcome. Studies included in this systematic review were quality scored by all authors using the Quality Assessment Tools for Case Series Studies and the Quality Assessment Tools for Observational Cohort and Cross-Sectional Studies, National Institute of Health, USA (https://www.nhlbi.nih.gov/health-topics/study-quality-assessment-tools, accessed on 1 February 2022). The studies were rated “good”, “fair”, or “poor” according to the estimated risk of bias. All authors performed the quality assessment, and any disagreements were solved by consensus. Due to the heterogeneity of the studies and the many different outcomes they used, a meta-analysis could not be performed.

## 3. Results

### 3.1. Study Selection

A total of 3996 titles and abstracts were identified using the two search strings presented in Table 1. After duplication screening, 493 duplicates were removed, resulting in 3503 unique hits. All titles and abstracts were screened, and subsequently, 169 full texts were read, returning 31 articles that met the abovementioned inclusion and exclusion criteria and, thus, were included in this systematic review. The inclusion and exclusion process is presented in Figure 1.

### 3.2. Study Description and Quality Assessment

Study characteristics for all included studies are presented in Table 2. All studies were either case reports (*n* = 17) or cohort studies (*n* = 14). In addition, an update of a case report was identified, and the sum of the two identified case reports was included [31,32]. The articles were published between 2015 and 2021. Twelve case reports were rated good, and the last five were rated as fair. Eleven cohort studies were rated good, two fair, and one poor.

In total, 80 patients representing 30 different histological subtypes were treated with targeted therapy based on a pre-specified target. Twenty-eight different drugs were used, targeting forty-one different targets. Fourteen patients received two or more different targeted agents in the same treatment course. Thirteen of the drugs were given only to one patient. The treatment effect on patients with advanced STS was measured by nine different treatment outcomes: complete response (CR), median time to progression, no evidence of disease (NED), overall survival (OS), partial response (PR), progression-free survival (PFS), progressive disease (PD), relapse-free survival, and stable disease (SD). In total, seven patients achieved CR, near CR, or NED. The patients achieving CR were treated as follows: Three patients received crizotinib targeting an NTRK1–KHDRBS1 fusion, ALK rearrangements, or an LMNA–NTRK1 fusion [32,33,34]; one patient received the MDM2 inhibitor, targeting an MDM2 amplification [11], and one patient received larotrectinib against a SPECC1L–NTRK3 fusion [35]. The two patients achieving NED received crizotinib against ALK expression and a SLC12A1–ROS1 fusion, respectively [36]. Only the MDM2 inhibitor, crizotinib, ceritinib, and ^90^Y-OTSA were given to eight or more patients.

#### 3.2.1. MDM2 Inhibitors

MDM2 inhibitors were given to 19 patients. All patients receiving MDM2 inhibitors were patients with liposarcoma and MDM2 amplification; furthermore, all patients had a treatment effect of SD or better with a treatment duration ranging from 4 to 83 months [11,13,37].

#### 3.2.2. Crizotinib

A total of 15 patients received crizotonib. The patients receiving crizotinib had NTRK1 fusions (*n* = 2), ALK rearrangements (*n* = 9), ROS1 mutations (*n* = 3), or MET mutations (*n* = 1) and achieved varying responses from PD to CR with a treatment duration ranging from three months to two years [11,32,34,38,39,40,41,42].

#### 3.2.3. Ceritinib

Ceritinib was given to eight patients. The patients receiving ceritinib had ALK rearrangements (*n* = 6), ROS1 amplifications (*n* = 1), or IGFR1 amplifications (*n* = 1) and achieved varying responses from PD to PR with a treatment duration of up to 24 months [11,41,43,44].

#### 3.2.4. ^90^Y-OTSA

Eight patients with FZD10 expression received the targeted radioactive drug ^90^Y-OTSA. Three patients received 370 MBq of these; one patient had SD, and two patients PD. Five patients received 1100 MBq; two had SD, and three had PD. One of the patients with SD receiving 1100 MB1 ^90^Y-OTSA also received a second injection that resulted in PFS for 21.4 weeks [45].

**Table 2 jpm-13-00730-t002:** Study characteristics.

First Author, Year	Study Design	Quality Assessment	Histological Subtype (n)	Genetic Alteration	Targeted Agent	Treatment Outcome
Arnaud-Coffin, 2020 [2]	Prospective cohort study	Good	Advanced STS			(PFS/OS)
			LMS (1)	AKT2 amplification	Everolimus	2.6/10.9 months
			MPNST (1)	ERBB2 mutation	Lapatinib	1.9/3.8 months
			Angiosarcoma (1)	FLT4 mutation	Pazopanib	3.1/10.7 months
			UPS (1)	AKT2 deletion	Everolimus	1.4/4.1 months
			GIST (3)	CDKN2A deletion	Palbociclib	0.8/4.9, 0.9/2.6, 3.7/22.9 months
Brian Dalton, 2017 [46]	Cohort study	Good	Advanced cancer			
			MPNST (1)	EGFR duplication	Afatinib	PD 2 months
			RMS (1)	FGFR1 mutation	Pazopanib	PD 4 months
Cecchini, 2018 [47]	Case report	Fair	High-grade sarcoma (1)	ATM frameshift mutation	Olaparib	PD 2 months
Chen, 2021 [33]	Case report	Good	Mesenchymal sarcoma (1)	NTRK1–KHDRBS1 fusion	Crizotinib	CR 40+ months
Dembla, 2018 [13]	Retrospective cohort study	Fair	LPS (6)	MDM2 amplification	MDM2 inhibitor	3 had PR, 2 had SD (15.7 and 4.7 months, respectively) 1 n/a
Elvin, 2017 [48]	Case report	Good	LMS (1)	CDKN2A deletion	Palbociclib	SD 4 months, radiological progression at 8 months
Forde, 2016 [31] Kinne, 2019 [32]	Case report Update	Fair	IMS (1)	ALKrearrangements	Crizotinib	CR 3 months, CR 164 weeks
Giraudet, 2018 [45]	Phase 1 cohort study	Good	SS (8)	FZD10	^90^Y-OTSA-101	
					3 received 370 MBq of ^90^Y	1 had SD, 2 had PD
					5 received 1110 MBq of ^90^Y	2 had SD, 3 had PD 1 with SD received a second injection resulting in PFS for 21.4 weeks
Groisberg, 2017 [11]	Cohort study	Good	Advanced sarcoma			
			Gliosarcoma (1)	BRAF V600E	Vemurafenib	PR 16 months
			DDLPS (1)	ROS1 amplification	Ceritinib	SD 5 months
			DDLPS (1)	MDM2 amplification	MDM2 inhibitor	PR 3 cycles
			GIST (1)	KIT +AKT Amplification	Imatinib, sunitinib, regorafenib, AKT inhibitor	PD x3
			LMS (2)	ROS1 mutation	Pazopanib +crizotinib	PR 22 cycles, SD 6 months, PD
			LMS (1)	PTEN deletion	PI3K inhibitor	PD
			Pleomorphic sarcoma (1)	MEMO1–ALK fusion	Ceritinib	PD after 4 cycles
			Myxoid LPS (1)	AKT1 mutation	AKT inhibitor	SD 1 cycle
			Spindle cell sarcoma (1)	KIAA1549–BRAF fusion	Sorafenib+ bevacizumab + temsirolimus	SD 11 cycles
			WDLPS (4)	MDM2 amplification	MDM2 inhibitor	SD 8 cycles, CR, SD 2 cycles, SD 23 months
Groisberg, 2020 [49]	Cohort study	Fair	Alveolar soft part sarcoma (1)	HGF amplification	Pazopanib + vorinostat	SD 28 months
Harttrampf, 2017 [50]	Prospective cohort study	Good	Paediatric advanced tumours Epithelioid sarcoma (1)	SMARCB1 deletion	Tazemetostat	PD 2 months
Ji, 2016 [51]	Case report	Good	Angiosarcoma (1)	VEGFR2	Apatinib	PFS 12 months
Jin, 2021 [52]	Cohort study	Good	STS			
			Inflammatory myofibroblastoma (1)	MAP2K1	Trametinib	RFS: 2 months
			LPS (1)	CDK4	Palbociclib	RFS: 4 months
			Fibrosarcoma (1)	COL1A1–PDGFB fusion	Imatinib	RFS: 10 months
			Clear cell sarcoma (1)	BRAF V600E	Vemurafenib	RFS: 21 months
Kato, 2018 [43]	Cohort study	Good	Advanced cancer			
			Desmoid tumour (1)	CTNNB1 mutation	Sorafenib + sulindac	SD, PFS 9.1+ months
			ESS (1)	CDKN2A mutations and FRS2 amplification	Palbociclib + lenvatinib + anastrozole + doxorubicin	SD, PFS 3.6+ months
			Myxofibrosarcoma (1)	IGFR1 amplification	Ceritinib	PD, PFS 1.8+ months
Kerr, 2021 [36]	Cohort study	Good	IMT (2)	ALK+ SLC12A1–ROS1 fusion	Crizotinib Crizotinib+ surgery	NED: 1.8 years NED: 2 years
Kyi, 2021 [44]	Case report	Good	IMT (1)	FN1–ALK fusion	Crizotinib→ ceritinib	SD 4 months→ SD 6 months
			IMT (1)	TNS1–ALK fusion	Crizotinib→ Alectinib→ Ceritinib → Lorlatinib	SD 3 months → SD 12 months → SD 2 months → PD 1 month
			Myofibroblastic sarcoma (1)	LBH–ALK fusion	Crizotinib→ Ceritinib	SD 30 months → SD 6 months
			IMT (1)	IGFBP5–ALK fusion	Ceritinib	PR 24+ months
Li, 2020 [42]	Case report	Fair	MPNST (1)	TJP1–ROS1 fusion	Crizotinib	SD 2 months, PD after 4 months
Mansfield, 2016 [41]	Case report	Good	IMS (1)	ALK+	Crizotinib → Ceritinib + surgery	PR 8 months→ progression → PR 18 months
Rabban, 2020 [35]	Case report	Good	Uterine sarcoma (1)	SPECC1L–NTRK3 fusion	Larotrectinib	CR 15+ months
Recine, 2021 [53]	Case report	Good	Spindle cell neoplasm (1)	TPM4–NTRK1 fusion	Larotrectinib	PR 19+ months
Seligson, 2021 [54]	Case report	Good	Small cell round tumour (1)	EWSR1–NFATc2 fusion	Everolimus + surgery + non-TT	SD 47 months
Seol, 2019 [15]	Prospective cohort study	Good	Advanced tumours			
			Uterine sarcoma (1)	AKT3 amplification	Everolimus	PR 5 months
Somaiah, 2018 [37]	Cohort study	Good	LPS (8)	MDM2 amplification	MDM2 inhibitor	Median time to progression: 23 months (95%-CI: 10–83 months)
Subbiah, 2015 [40]	Case report	Fair	IMS (1)	DCTN1–ALK fusion	Crizotinib + pazopanib	PR 6+ months
Subbiah, 2020 [55]	Cohort study	Good	Advanced tumours Clear cell sarcoma (1)	Placental cadherin	^90^Y-FF-21101 mAb 25 mCi/m^2^	SD 50 weeks
Valenciaga, 2021 [56]	Case report	Good	Pleomorphic LPS (1)	IQGAP–NTRK3 fusion	Entrectinib → Pazopanib + radiation → Larotrectinib	PD 4 cycles → PD 3 months → SD 18+ months
Walsh, 2021 [57]	Case report	Fair	IMT (1)	LRRFIP1–ALK fusion	Alectinib	PR 19 months
Weidenbusch, 2018 [39]	Cohort study	Poor	Paediatric sarcoma			
			RMS (1)	MET and FGFR1 mutation	Crizotinib + ponatinib	SD 7 months
			SS (1)	FGFR1 and EGFR mutation	Ponatinib + gefitinib	PD
Wu, 2021 [58]	Case report	Good	Primary pulmonary artery sarcoma (1)	Loss of ATM and H2AX	Olaparib	PR 2 months
Yang, 2018 [38]	Case report	Good	Myofibroblastic sarcoma (1)	ALK mutation	Crizotinib + bevacizumab	PFS 3 months
Zhou, 2018 [34]	Case report	Good	UPS (1)	LMNA–NTRK1 fusion	Crizotinib	Near-CR 18+ months

Abbreviations: CR: complete response, DDLPS: dedifferentiated liposarcoma, ESS: endometrial stromal sarcoma, GIST: gastrointestinal stromal tumour, IMS: inflammatory myofibroblastic sarcoma, IMT: inflammatory myofibroblastic tumour, LMS: leiomyosarcoma, LPS: liposarcoma, mAb: monoclonal antibody, MPNST: malignant peripheral nerve sheath tumour, n/a: not applicable, NED: no evidence of disease, OS: overall survival, PD: progressive disease, PFS: progression-free survival, PR: partial response, RFS: relapse-free survival, RMS: rhabdomyosarcoma, SD: stable disease, SS: synovial sarcoma, TT: targeted treatment, UPS: undifferentiated pleomorphic sarcoma, WDLPS: well-differentiated liposarcoma.

## 4. Discussion

Advanced sarcoma is a rare but severe disease, and better treatment options are needed. We have conducted a systematic review to cover targeted treatment for patients suffering from advanced STS with a known genetic alteration.

The use of targeted treatment in STS is still on an individual and experimental level. Most of the studies included in this systematic review were case reports or cohort studies where only a few individuals suffering from advanced STS were included. Summarised, 80 patients received 28 different targeted drugs. Thirty different subtypes of STS received targeted treatment, and among them, forty-one different genetic alterations were identified. Thus, several targeted agents were administered to patients with different kinds of advanced STS but with highly variable results.

Several studies have shown that there are many different genetic alterations in STS. Groisberg et al. found that 93 out of 102 advanced STS patients had at least one genetic alteration, where the most frequent were found in TP53, CDK4, and MDM2, and 61% had a potentially targetable alteration. However, only 16% of the patients received a targeted drug [11]. Similar observations were found by Lucchesi et al., where 84% of the 584 included STS patients enharboured a genetic alteration. Furthermore, the most frequent mutations were observed in TP53, MDM2, and CDK4, and 41% of the patients had at least one potentially targetable alteration [18]. Gusho et al. also found a high frequency of genetic alterations with at least 1 genetic alteration in 122 out of 136 samples. The most frequent mutations observed were in TP53, CDKN2A/B, and RB1, and 47% had a potentially targetable mutation [19].

In this review, all patients receiving MDM2 inhibitors had liposarcomas, with a treatment response of SD or better, including one CR [11,13,37]. MDM2 amplification is a common genetic alteration in STS. Three studies have shown a 5%, 22%, and 40% prevalence of STS [2,11,13]. In specific subtypes of STS, the prevalence is even higher. In well-differentiated and dedifferentiated liposarcoma, the amplification of 12q13-15 is common, resulting in the amplification of the MDM2 gene. Studies have shown that over 90% of these sarcomas have MDM2 amplification [13,37]. Because of the high prevalence of MDM2 alterations in STS, especially in liposarcoma, and the reports with high clinical effects, targeting this specific alteration must be considered.

Crizotinib was given to patients with mutations in ALK, ROS1, MET, or NTRK1. There were varying treatment responses, but three out of nine patients had CR or near-CR, and two patients had NED; therefore, they had an overall survival benefit. One of the patients with CR had ALK gene rearrangements, and the other had NTRK1–KHDRBS1 fusion. The patient with a near-CR response had an LMNA–NTRK1 fusion. The patients with NED had high ALK expression and a SLC12A1–ROS1 fusion [32,33,34,36].

Owing to the aim of this study, only patients receiving treatments targeting a known specific target were included. Therefore, only seven of the included patients received pazopanib, a multi-target tyrosine kinase inhibitor; however, the drug is given to far more patients suffering from advanced STS. Even though studies exist on pazopanib in STS, these studies were not included as they did not have a specific pathological proven target before treatment with pazopanib. Many of these studies showed a treatment effect on STS without prior identification of a specific genetic alteration.

MDM2 inhibitors, crizotinib, ceritinib, and ^90^Y-OTSA were given to most patients. However, the population receiving targeted therapy is so tiny that it is difficult to conclude the overall survival effect of the targeted agents. Furthermore, nine different treatment outcomes were used to measure the effectiveness of the drugs, making it challenging to conduct a meta-analysis to investigate the potential survival effect of agents on advanced STS.

Most of the included patients were previously treated with curative-intended treatments, and many had one or more treatment courses with chemotherapy. Furthermore, targeted agents were tried as a last resort for patients with advanced STS. Maybe, if the targeted therapy had been introduced to the patients before relapse occurred, the patients might have had a more substantial benefit from the treatment and more prolonged survival. It would have been interesting to compare targeted treatment against chemotherapy as the first-line treatment for advanced STS to see if there is a difference. Larger studies are needed to investigate the potential effect of targeted therapy on STS, for example, case/control studies, where targeted treatment is compared to chemotherapy and ultimately randomised controlled trials.

The strengths of this systematic review are the comprehensive systematic review of the available literature on targeted treatment of STS in two major medical databases. Broad search strings and in- and exclusion criteria were used to avoid missing relevant articles. Two authors screened the literature, and all authors validated the in- and exclusion criteria. Nonetheless, this review also has its limitations. All the studies included are either case reports or cohort studies, where only a few patients have advanced STS. The studies used many different measures of outcome. One of the exclusion criteria was non-English articles, and it is possible that other studies concerning the same topic were written in other languages. Thus, language bias cannot be excluded.

## 5. Conclusions

During the last decades, many genetic alterations in STS have been identified, enabling the use of targeted treatment. This systematic review revealed many studies regarding genetic alterations and targeted treatment in advanced STS. Twenty-eight targeted agents have currently been tried in advanced STS. However, most articles are case reports and cohort studies representing only a few patients with advanced STS. Studies comparing targeted treatment to chemotherapy in case/control studies and ultimately randomised controlled trials can make it easier to investigate the potential survival benefit that targeted agents can provide to patients suffering from advanced STS.

## Figures and Tables

**Figure 1 jpm-13-00730-f001:**
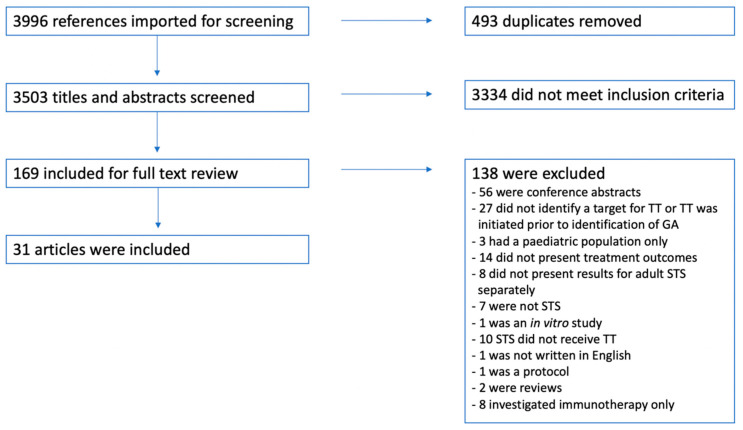
A flow chart presenting the in- and exclusion process. Abbreviations: GA: genetic alteration, STS: soft-tissue sarcoma, and TT: targeted treatment.

**Table 1 jpm-13-00730-t001:** Presentation of the search strings.

Database	Search Strings	Number of Results
PubMed	(“Sarcoma”[MeSH Terms] OR “soft tissue sarcoma*”[Text Word] OR “soft tissue neoplasm*”[Text Word]) AND (“adult*”[Text Word] OR “Adult”[MeSH Terms]) AND (“Molecular Targeted Therapy”[MeSH Terms] OR “Immunotherapy”[MeSH Terms] OR “molecular targeted therapy*”[Text Word] OR “immunotherapy*”[Text Word] OR “targeted therapy*”[Text Word])	846
Embase	(‘sarcoma’/exp OR sarcoma*:ab,kw,ti OR ‘soft tissue sarcoma*’:ab,kw,ti OR ‘soft tissue neoplasm*’:ab,kw,ti) AND (‘adult’/exp OR adult*:ab,kw,ti) AND (‘molecularly targeted therapy’/exp OR ‘molecularly targeted therapy*’:ab,kw,ti OR ‘immunotherapy’/exp OR immunotherapy*:ab,kw,ti OR ‘targeted therapy*’:ab,kw,ti) AND [humans]/lim AND [english]/lim	3150

## Data Availability

Not applicable.

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
