# Peer review of "Targeted Treatment of Soft-Tissue Sarcoma"

_jpm, 2023, doi:10.3390/jpm13050730_

Round 1

Reviewer 1 Report

Dear editor,

Thanks for giving this opportunity to review the manuscript entitled “ Targeted treatment of soft-tissue sarcoma” by Anne Iren Riskjell, et al. The authors aimed to generate an overview of the clinical experiences with targeted treatments based on a pre-specified target in patients with advanced STS. From the literature review, 80 patients with advanced STS had a known pre-specified genetic alteration were treated with twenty-eight targeted agents. Patients received MDM2 inhibitor achieved a treatment response of stable disease (SD) or better with a treatment duration of 4 to 83 months. Therefore, the authors concluded that many targeted agents are available that can precisely target specific genetic alterations in advanced STS. The MDM2 inhibitor has shown promising results.

Although most articles included in this paper are case reports and cohort studies, this systematic review for the first time to cover treatment options against specific genetic alterations in STS exists. Hopefully, future randomized controlled trials comparing targeted treatment to chemotherapy can better show the potential survival benefit that targeted agents can provide to patients suffering from advanced STS.

It’s a well-written paper. I don’t have specific comments regarding the manuscript. We may recommend this manuscript to be published.

Author Response

Dear editors and reviewers,

Thank you for your valuable comment on our research article: "Targeted treatment of soft tissue sarcoma", which we believe has improved our manuscript substantially.

The responses to the commentary are shown below.

Kind regards.

Ninna Aggerholm-Pedersen

MD, PhD, MSc, Associate Professor

on behalf of all the authors

Reviewer 1

Question:

Thanks for giving this opportunity to review the manuscript entitled “ Targeted treatment of soft-tissue sarcoma” by Anne Iren Riskjell, et al. The authors aimed to generate an overview of the clinical experiences with targeted treatments based on a pre-specified target in patients with advanced STS. From the literature review, 80 patients with advanced STS had a known pre-specified genetic alteration were treated with twenty-eight targeted agents. Patients received MDM2 inhibitor achieved a treatment response of stable disease (SD) or better with a treatment duration of 4 to 83 months. Therefore, the authors concluded that many targeted agents are available that can precisely target specific genetic alterations in advanced STS. The MDM2 inhibitor has shown promising results.

Although most articles included in this paper are case reports and cohort studies, this systematic review for the first time to cover treatment options against specific genetic alterations in STS exists. Hopefully, future randomized controlled trials comparing targeted treatment to chemotherapy can better show the potential survival benefit that targeted agents can provide to patients suffering from advanced STS. It’s a well-written paper. I don’t have specific comments regarding the manuscript. We may recommend this manuscript to be published.

Answer:

We thank reviewer 1 for the nice comment on our paper.

Reviewer 2 Report

In the review “Targeted treatment of soft-tissue sarcoma” the authors performed a wide literature screening and generated an overview of the targeted treatments employed against advanced soft-tissue sarcomas. The criteria used for the selection of papers included in the study are very clear, the paper is well written and the results describe the effect of different targeted agents used in the treatment of a wide number of patients affected by soft-tissue sarcomas. The table includes several studies and collects all the therapies used, the hytsological and genetical classification of each tumor and the correlation with patients prognosis. However, since the topic of the manuscript is the targeted treatment, maybe the studies could be grouped based on the targeted agent employed in each work. This will help the reader to better focus on the role and on the efficacy of every single inhibitor and better underline the most used and effective targeted agents against the various subtypes of advanced soft-tissue sarcomas. Furthermore, a figure or a final scheme summarizing the main inhibitors or the most important findings of the work could make the results more appreciable.

Author Response

Dear editors and reviewers,

Thank you for your valuable comment on our research article: "Targeted treatment of soft tissue sarcoma", which we believe has improved our manuscript substantially.

The responses to the commentary are shown below.

Kind regards.

Ninna Aggerholm-Pedersen

MD, PhD, MSc, Associate Professor

on behalf of all the authors

Reviewer 2

Question 1:

In the review “Targeted treatment of soft-tissue sarcoma” the authors performed a wide literature screening and generated an overview of the targeted treatments employed against advanced soft-tissue sarcomas. The criteria used for the selection of papers included in the study are very clear, the paper is well written and the results describe the effect of different targeted agents used in the treatment of a wide number of patients affected by soft-tissue sarcomas. The table includes several studies and collects all the therapies used, the hytsological and genetical classification of each tumor and the correlation with patients prognosis. However, since the topic of the manuscript is the targeted treatment, maybe the studies could be grouped based on the targeted agent employed in each work. This will help the reader to better focus on the role and on the efficacy of every single inhibitor and better underline the most used and effective targeted agents against the various subtypes of advanced soft-tissue sarcomas.

Furthermore, a figure or a final scheme summarizing the main inhibitors or the most important findings of the work could make the results more appreciable.

Answer:

Thank you for the constructive comments. We agree that presenting the results grouped by targeted treatments would improve the manuscript and underline the focus of our study substantially. Therefore, the results section has now been rewritten.

Reviewer 3 Report

Dear Authors, 

I read with attention your paper, but i can not support the publication for the following reasons:

- the review done is mostly across case reports and few brief reports not touching in deep well known drugs

- clinical setting is poorly described and argumented

- no data were reported on how decision of treatments was done, in which disease state was admnistered and toxicity data

- no sentences focused on precision medicine and genetic screen

- the length of the text is too short to be a review

- the focus of this review is incomprehensible

Author Response

Dear editors and reviewers,

Thank you for your valuable comment on our research article: "Targeted treatment of soft tissue sarcoma", which we believe has improved our manuscript substantially.

The responses to the commentary are shown below.

Kind regards.

Ninna Aggerholm-Pedersen

MD, PhD, MSc, Associate Professor

on behalf of all the authors

Reviewer 3

Question 1:

- the review done is mostly across case reports and few brief reports not touching in deep well known drugs.

Answer:

The reviewer is right regarding the content of this paper. The systematic review aims to give an overview of all not known targeted drugs used in sarcoma. The most used targeted drug in sarcoma is imatinib for Gastrointestinal stromal tumours (GIST). GIST patients are not the focus of this article.

It is also well known that PDL and PDL1 are not markers for the effect of checkpoint inhibitors and are therefore not included as well; this has been discussed in the discussion section on page xx, line xx

“Patients with Gastrointestinal stomal Tumours (GIST) treated with tyrosine kinase inhibitors (TKIs) are not included in the systemic review even though they are treated with TKIs targeting different tyrosine kinases. The evidence for this treatment is well known, and for a systematic review of treating GIST, we refer to a systematic review published by Brinch et al.[16]. The multitargeted drug Pazopanib, a tyrosine kinase inhibitor targeting VEGFR, platelet-derived growth factor receptor and c-kit are not included, as the treatment is given to many sarcoma patients with genetic testing with good clinical responses[17-23]. The same is true for regorafenib[24,25]

The use of pazopanib and regorafenib is also a well know effective drug in sarcoma. This multitarget tyrosine kinase inhibitor is not included as well. The following has been added to the material and method section (Page x, line x) to clarify this point. The studies included in this systematic review are selected for treatment given based on the results from comprehensive genetic testing. Therefore the following has been added to the introduction page x, line x

“The studies included in this systematic review are selected for treatment given, based on the results from comprehensive genetic testing.”

Question 2:

- clinical setting is poorly described and argumented

Answer:

We agree with the reviewer that the background for this systematic review is poorly described; therefore, a more comprehensive introduction has been added (page 1, line 43)

“STS are primarily driven by a fusion of genes rather than mutations. However, several studies have shown some genetic alterations in STS. Cancer Genome Atlas (TCGA) described the molecular landscape of 206 adult STS, representing six subtypes of STS, including synovial sarcoma (SS), liposarcoma (LPS), leiomyosarcoma (LMS), malignant peripheral nerve sheath sarcoma (MPNST), myxofibrosarcoma and undifferentiated sarcoma. They found that most sarcomas are characterised by copy number alterations (CNA) and a low tumour mutation burden (TMB) [8]. TMB is usually categorised into three categories: low (1-5 mutations/Mb), intermediate (6-19 mutations/Mb) and high (≥ 20 mutations/Mb) [9]. The average TMB in TCGA was low for sarcoma, with an average of 1.06 mutations/Mb. Other mutations represent a few highly recurrent genes. The most frequently mutated genes in the database were TP53 (n=69), ATRX (n=31), RB1 (n=18), PCLO (n=9), FAT1 (n=6), NF1 (N=6), PRKDC (n=6) and LRP1B (n=6). The most frequent amplifications were seen in MDM2 (n=46), FRS2 (n=45), CDK4 (n=44), HMG2A (n=36) and PTPRB (n=33). The most frequent deletions were observed in CDKN2A (n=22), CDKN2B (n=22), RB1 (n=22), CYSLTR2 (n=18) and TP53 (n=16) [8]. In a study investigating the mismatch-repair (MMR) status of 304 sarcomas, seven were found to be MMR-deficient. MMR-deficient sarcomas had a significantly higher TMB than MMR-proficient, with an average of 16.95 mutations/Mb and 4.56 mutations/Mb, respectively[10].

Arnaud-Coffin et al. investigated 158 patients with advanced STS by genetic profiling. They found 289 relevant genetic alterations in 149 genes. The most frequent alterations were TP53 (n=36), RB1 (n=22), CDKN2A (n=17), CDK4 (n=9), MDM2 (n=8) and PTEN (n=7). The alterations in CDK4 and MDM2 were amplifications only [2]. In a study including 102 sarcoma patients, the most frequent alterations were also TP53 (n=32), CDK4 (n=24), MDM2 (n=22), RB1 (n=19) and CDKN2A/B (n=14) [11]. In a study investigating the molecular characterisation of fourteen adult STS, the most frequently altered genes were FRGB1 (n=8) and CDC27 (n=6). TP53, ARTX and PTEN were mutated in three cases. Median TMB were low (2.38 mutations/Mb) [12]. Dembla et al. found the prevalence of MDM2 amplification in 13/33 sarcoma patients [13]. Lin et al. included 301 with uterine sarcomas representing many subtypes of STS. Here they found that nineteen were SMARC4-deficient. In the SMARC4-deficient cohort, they performed next-generation sequencing (NGS) on sixteen patients. Average TMB was low (1.7 mutations/Mb) and they found mutations in TP53 (n=2), RB1 (n=1), CTNNB1 (n=1) and ZNF703 (n=1) [14]. Seol et al. also analysed five patients with uterine sarcoma with NGS. In one patient, they found amplification of AKT3, BRAF and EGFR. Another patient had amplification of PDGFRB [15]. Li et al. investigated forty BCOR-rearranged uterine sarcomas. Among these patients, thirty-eight had ZC3H7B-BCOR fusion. They also found amplification of MDM2 (n=18), FRS2 (n=16), CKD4 (n=15), PDGFRA (n= 3), KDR (n= 2), ERBB3 (n= 2) and KIT (n=1). Loss of CDKN2A and CDKN2B were observed in eleven and seven cases, respectively. They also found inactivating mutations in TP53 (n=4), PTCH1 (n=2), NF1 (n=2) and NF2 (n=1). In addition to BCOR, other rearrangements were in the HMGA2 and NCOR2 genes with 6 and 2 cases, respectively. Thirty-nine patients had a low TMB, and one had intermediate. In addition to the BCOR-cohort, they also investigated a cohort consisting of fifteen patients with BCOR internal tandem duplication. Here none of the cases had MDM2 and CDK4 amplification. Three of the cases had CDKN2A/B loss and mutations in STAG2 (n=2), PASK (n=2) and ARID1A (n=2) [16]. Thirteen patients with renal sarcoma had a low TMB with a mean of 3.5 mutations/Mb. Amplification of KIT and PDGFRA were observed in four patients. Three cases experienced a loss of CDKN2A/B. Genetic alterations were also found in TP53 (n=4), NF1 (n=3), MLL2 (n=2). One case had a fusion of STAT6-NAB2 [17].

In STS, the frequency of genetic alterations is between 84-91%, with the most frequently altered genes being TP53, ATRX, RB1, PTEN, MDM2, CDK4 and CDKN2A/ B [2,11,13,16-20]. Different genetic alterations can enable targeted therapy in patients suffering from advanced STS. However, a lot is currently known about the genetic landscape of sarcoma, but no overview covering treatment options against specific genetic alterations in STS exists. This systematic review aims to generate an overview of the clinical experiences with targeted treatments based on a pre-specified target in patients with STS.”

Question 3:

- no data were reported on how decision of treatments was done, in which disease state was admnistered and toxicity data

Answer:

This systematic review aims to overview targeted treatment based on results from gene sequencing where genetic alterations have been found. Therefore, the patient treated in all has a local advanced or metastatic disease, and most patients have been treated with standard chemotherapy before the genetic testing. This has been clarified on page 3, line 116, in the introduction.

. Genetic testing is primarily done in sarcoma patients with locally advanced or metastatic disease, who have been treated with known standard palliative treatment.”.

It was not the objective to report on toxicity or how to administrate the drug; as the reviewer pointed out, this is primarily based on case reports or case series.

Question 4:

- no sentences focused on precision medicine and genetic screen

Answer:

To clarify, the focus of this systematic review is precision medicine; the following has been added to the introduction (page 3, line 114). The genetic screening has been addressed; see the answer to question 2.

This study focuses on the outcome after target treatment of rare genetic alterations in sarcoma patients, not the genetic alteration by themselves or already effective treatments. Genetic testing is primarily done in sarcoma patients with locally advanced or metastatic diseases, who have been treated with known standard palliative treatment.”.

Question 5:

- the length of the text is too short to be a review

Answer:

The length of the text has been prolonged to more than 4000 words.

Question 6:

- the focus of this review is incomprehensible

Answer:

The focus of this article has been clarified, and a comprehensive introduction about the genetic landscape of sarcoma has been added to the introduction.

Round 2

Reviewer 3 Report

Dear Authors, 

I read with attention your paper, but i can not support the publication 

Sincerely

Author Response

The manuscript has now been reviewed by a professional team regarding the English language. Corrections are marked in the reviewed manuscript.

The e-mail address for the first author has been changed to annrij@rm.dk

Reviewer 3 does not support the paper's publication, whereas reviewers 1 and 2 find the systematic review well-written and relevant. In response to reviewer 3 we have the following comment.

The paper is the first to collect information on targeted treatment for specific genetic alterations in sarcoma. Sarcoma is a rare cancer with about 100 different histological subtypes; therefore, a review of the rare targets and effects of targeted treatment is very useful for clinicians when guiding patient treatment undergoing next-generation sequencing. Moreover, we, as clinicians, find the review suitable for publication.
